# Stress-Related Responses to Alternations between Repetitive Physical Work and Cognitive Tasks of Different Difficulties

**DOI:** 10.3390/ijerph17228509

**Published:** 2020-11-17

**Authors:** Susanna Mixter, Svend Erik Mathiassen, Petra Lindfors, Kent Dimberg, Helena Jahncke, Eugene Lyskov, David M. Hallman

**Affiliations:** 1Centre for Musculoskeletal Research, Department of Occupational Health Sciences and Psychology, University of Gävle, 801 76 Gävle, Sweden; SvendErik.Mathiassen@hig.se (S.E.M.); helena.jahncke@afaforsakring.se (H.J.); eugene.lyskov@hig.se (E.L.); david.hallman@hig.se (D.M.H.); 2Department of Psychology, Stockholm University, 114 19 Stockholm, Sweden; pls@psychology.su.se; 3Department of Electrical Engineering Mathematics and Science, University of Gävle, 801 76 Gävle, Sweden; kent.dimberg@hig.se

**Keywords:** recovery, mental task, physical task, women, repetitive work, job rotation

## Abstract

Alternating between physical and cognitive tasks has been proposed as an alternative in job rotation, allowing workers to recover from the physical work while still being productive. However, effects of such alternations on stress have not been investigated. This controlled experiment aimed at determining the extent to which stress-related responses develop during alternating physical and cognitive work, and to determine the extent to which cognitive task (CT) difficulty influences these responses. Fifteen women performed three sessions of 10 consecutive work bouts each including a seven-minute repetitive physical task (pipetting) and a three-minute CT (*n*-back) at one of three difficulty levels. Stress was assessed in terms of changes in heart rate variability, blood pressure, salivary alpha-amylase, salivary cortisol, perceived stress, and cognitive performance. The work session did not result in any marked stress response, and CT difficulty did not significantly influence stress, apart from alpha-amylase being higher at the easiest CT (F = 5.34, *p* = 0.02). Thus, according to our results, alternating between repetitive physical tasks and cognitive tasks may be a feasible alternative to classic job rotation between physical tasks only, even if the cognitive task is quite difficult. Future studies should address possible effects of the temporal pattern of alternations, and combine even other occupationally relevant tasks, preferably for extended periods of time.

## 1. Introduction

Increased variation in biomechanical exposure has been proposed by authorities and researchers as a remedy for reducing risks of musculoskeletal disorders (MSD), particularly in jobs dominated by repetitive and constrained tasks [1,2,3,4,5]. Variation can be modified by changing rest allowances at work [5], and the effectiveness of rest in providing recovery from muscle fatigue has, indeed, been confirmed in numerous studies of isometric, isotonic contractions [6,7,8]. However, extending or restructuring rest breaks appears to have limited effect on fatigue and discomfort in occupational settings [9]. One explanation may be that rest allowances can be implemented only to a limited extent because breaks are essentially non-productive time [5,7]. Job rotation is another way of increasing physical variation, but studies so-far show inconclusive results regarding whether job rotation is, indeed, effective in increasing variation and reducing musculoskeletal complaints [10,11,12].

Job rotation schemes often consist of alternations between different physically demanding work tasks. An alternative involves including a (productive) mental work task with little biomechanical demands in an otherwise physical work, one example being workers at an assembly line performing administrative work in between bouts of physical work [13]. This would allow workers to recover from physical fatigue, as with a rest break, however without losing productive time. Alternations between physical and mental tasks occur already in many occupations, including retail, service-jobs, and certain industries, and may even be preferred by workers, compared to performing either physical or mental tasks only [14].

Alternations between activation and recovery are parts of an adaptive stress response, as emphasized in the allostatic load model [15]. In contrast, prolonged and persistent activation of stress regulatory systems can result in allostatic load, and potentially increase the risk of MSD [16,17,18,19] and cardiovascular disease (CVD) [20]. In a biopsychosocial view on stress and MSD, workers performing repetitive and simple tasks, are at greater risk for developing MSD, in particular if the psychosocial work environment is unfavorable as well [21]. Inadequate regulation of autonomic nervous system (ANS) activity in terms of a reduced vagal tone, as reflected in lower heart rate variability (HRV) [22,23,24], has been proposed as a mechanism linking work stress and CVD. Moreover, prolonged stress is associated with an impaired hypothalamic–pituitary–adrenal (HPA)-axis response and an inability to respond adequately to everyday stressors [25]. However, sufficient recovery is considered to alleviate negative effects and to promote health [26,27]. Keeping with the allostatic load model, alternating between physical tasks leading to fatigue and mental tasks offering recovery may be a viable option when designing healthy jobs.

Experimental studies suggest that a high mental workload, both in its own right and if it occurs concurrently with a physical task, imposes a marked activation of stress regulatory systems, compared to rest or a light mental workload [28,29,30,31,32,33,34]. However, it has seldom been investigated whether stress is an issue when a mental task, difficult or not, alternates with a physical task, even though this is relevant in the context of occupational applications. Some studies have addressed alternating physical and mental tasks in relation to fatigue and performance [35,36,37], even if few have addressed occupationally relevant tasks [38,39]. Only one study has addressed the influence of mental task difficulty on stress-related responses [39]. In that study, recordings of HRV and blood pressure among university students alternating between a repetitive physical task and a cognitive task did not suggest any stress reaction, regardless of cognitive task difficulty. However, a more comprehensive selection of stress indicators is needed to reflect both sympathetic nervous activation and HPA-axis response (e.g., salivary alpha amylase (sAA) and cortisol [40,41,42], and parasympathetic activation (HRV) [43].

Previous studies of alternations have been performed on men, although women tend to report more MSDs and work-related stress [44,45,46,47]. As an example, a cross-sectional study of a large sample from the Dutch working population found that the prevalence of neck complaints was 17.7 percent among women, but only 10.7 percent among men [45]. Thus, studies specifically addressing stress among women are needed, particularly for occupational tasks that women may perform to a larger extent than men, such as repetitive, short-cycle work [47].

Thus, the aim of the present study was to determine the extent to which stress develops during alternating physical and cognitive work in a population of women. We hypothesized that stress would increase over time and be more pronounced with increasing difficulty of the cognitive task. The study is part of a larger controlled experiment investigating alternating physical and cognitive tasks in women [38].

## 2. Materials and Methods

### 2.1. Participants

Fifteen women were recruited through advertisements at the University Campus (mean age: 26.5 years (SD 4.7); height: 1.67 m (SD 0.06); weight: 66 kg (SD 4.9); BMI 23.7 kg·m^−2^ (SD 1.9)). Inclusion criteria were previous experience of pipetting, age between 20 and 50 years, and right-handedness (assessed by the Edinburgh Inventory [48]). Exclusion criteria were pregnancy, any chronic disease, pain, or previous trauma to the neck or back. None of the participants used cigarettes or snuff on a daily basis. This research complied with the Declaration of Helsinki and was approved by the Regional Ethical Review Board in Uppsala, Sweden (Ref. No. 2014/002). Informed consent was obtained from each participant.

### 2.2. Study Design

Participants visited the lab for one training session (approximately 1 h) followed by three experimental sessions (approximately 4 h each), interspersed by between 3 to 7 days. All sessions for a particular participant (except two) were carried out at the same time of the day (either 8.30 AM to 12.30 PM; *n* = 9, or 1.30 PM to 5.30 PM; *n* = 4). The experiments were performed in a sound insulated room at an ambient temperature of 21–22° Celsius; air flow and humidity kept within recommended values; and sufficient luminance to perform the work tasks without visual hindrances. A window was covered with a blind to block external stimuli.

The experimental protocol consisted of a pre-test battery, ten 10-min bouts of a 7-min physical repetitive task (pipetting) followed by a 3-min cognitive task (*n*-back), and after that a post-test battery (Figure 1). The physical task was identical in all three sessions, while the difficulty level of the cognitive task (CT) was specific to the particular session; i.e., easy, moderate or difficult. The order of CT difficulty between sessions was counterbalanced to avoid any order effect of experimental days, and participants were not informed about the difficulty level before the session. Participants were instructed to avoid intense physical exercise for 24 h before each study session; avoid nicotine consumption during the hour before each experiment; and avoid eating, drinking, and brushing their teeth for 30 min before the experiments.

### 2.3. Physical Task

The physical task consisted of pipetting at a customized workstation. This pipetting task has been used in previous studies as a model of repetitive work [49,50] and has been demonstrated to lead to considerable fatigue when performed without interruption [51]. In other studies, pipetting has been reported to imply biomechanical exposures that likely lead to increased risk of MSD in the long term [52,53,54]. The present task included aspiring a pre-set volume of liquid sequentially from a pickup tube (Ø 20 mm) and transferring it to one of four smaller target tubes (Ø 6 mm) (Figure 2). A led lamp lit the target tube to which the liquid was to be dispensed. Thus, each pipetting cycle (2.8 s, guided by a metronome) included aspiring and dispensing liquid, at a pace corresponding to 100 MTM (Method Time Measurement System) [55]. The workstation was adjusted according to standard ergonomic guidelines; the height of the table was aligned with the elbow when the participant’s arm rested on the table at a 90° angle, and the height of the chair was set so that the knee angle was 90°. The chair and the table were arranged so that the participant’s wrist was right above the pickup tube (Figure 2) when she reached out for that tube.

Participants were strapped to the chair to avoid large variations in posture, which could introduce noise in the eventual data.

### 2.4. Cognitive Task (CT)

The CT was a commonly used working memory test, n-back [56], requiring information to be both maintained and updated in response to a stimulus [57,58]. The n-back task in the form used by us consisted of presenting a black letter (one of seven consonants) to the participant for 2000 ms on a white computer screen, followed by a blank screen for 500 ms. Participants were then instructed to press a button when the letter shown was identical to the previous letter (n-back 1), or the letter two steps back (*n*-back 2), or three steps back (*n*-back 3), with these “delays” representing an increased level of difficulty. In planning the present study, we conducted a pilot experiment to determine which n-back levels to include. Results from those experiments showed that n-back 1 (easy), 2 (moderate) and 3 (difficult) were best suited for the eventual experiment. As intended, performance decreased between n-backs 1 and 2, and then further between n-backs 2 and 3. A further increase in difficulty, to n-back 4, resulted in a marked further drop in performance, to the extent that several participants gave up. One 3-min bout of n-back in the eventual experiment included, in total, 72 letter presentations, with 15 randomly distributed letters being “correct”, i.e., requiring a reaction. The easy, moderate, and difficult task resulted in, on average, 14.0, 12.8, and 8.5 correct positive answers [38], confirming that the *n*-back levels differed markedly in difficulty (*p* < 0.001, ηp^2^ = 0.70), just as found in the pilot experiment and also reported by other studies [58]. We considered the n-back 1 as a control condition, since it was very easy, yet requiring participants to focus [39,57] and thus be in a more controlled mental state of relevance to the two other CT difficulties than if we had used passive, uncontrolled rest. To diminish physical load on the right body side, participants were instructed to press the button with their left (non-dominant) hand. Performance in each n-back condition was measured as the number of *correct positive answers* and *false positive answers* (i.e., pressing the button despite no match).

### 2.5. Familiarization and Training Trial

On a separate day, participants performed a one-hour familiarization and training trial, during which baseline data were collected. Before entering the experiment, participants had to complete at least 70 cycles of pipetting without any mistakes [49,59]. Additionally, participants performed a 10-min CT practice, including all difficulty levels.

### 2.6. Pre- and Post-Test Battery

At the start of an experimental session, the examiner asked participants about sleep quality, engagement in heavy physical activity 24 h prior to the experiment and consumption of drinks, food, and tobacco. The experiment was discontinued if a participant reported poor sleep quality; engagement in high-intensity physical activity 24 h prior to the experiment; or consumption of drinks, food, and tobacco within 30 min prior to the experiment. However, no experiments were discontinued due to this reason.

Participants then rated perceived stress on the Borg CR-10 scale [60,61] and provided a saliva sample (Figure 1), after which they completed a pre-test battery with an initial rest for five minutes, blood pressure (BP) measurements, and a five-minute training session on the “CT of the day” to avert any training effects during the experiment. Ratings of stress, BP measurement, and saliva sampling were repeated just before the 10 work bouts with alternating tasks (Figure 1). After the ten work bouts, a post-test battery was completed, with ratings of stress, BP measurement, and saliva sampling.

### 2.7. Physiological and Psychophysical Measurements

#### 2.7.1. Ratings

During the last minute of each pipetting bout and during a 30 s break after each CT bout, participants rated their perceived stress on a Borg CR-10 scale [60,61]. The CR-10 scale is a validated, general intensity scale which has been used to measure perceived stress also in previous studies [62]. Participants communicated their ratings verbally to the examiner.

#### 2.7.2. Electrocardiography, Heart Rate Variability, and Blood Pressure

Electrocardiography (ECG) was recorded throughout the experimental session from a standard two-led configuration using pre-gelled Ag/AgC1 electrodes (Ambu Blue Sensor VLC, Penang, Malaysia), pre-amplified (Noraxon, MyoSystem 1400A; gain 500), and sampled at a frequency of 2000 Hz using a customary digital 0.5–200 Hz band pass filter (Platon version 8.1). Further amplification was performed (Brownlee Precision Model 440 Instrumentation Amplifier) with a gain of 5. Data files were stored on a PC and imported to the Spike software (Spike 2, Cambridge Electronic Design, Cambridge, UK) for off-line processing. The time series of R–R intervals were visually inspected for artefacts, which were replaced using linear interpolation. HRV indices reflecting parasympathetic activity [43] were determined both in the time domain (i.e., rMSSD; root mean square of the successive differences between R–R intervals) and in the frequency domain (HF; high frequency spectral power: 0.15–0.4 Hz) according to previous recommendations [63]. These HRV indices are reliable markers of autonomic activity during light repetitive work [64]. Power spectral density of HRV was assessed using the Fast Fourier Transform. R–R intervals and HRV indices were obtained in consecutive, non-overlapping 2-min windows.

Systolic and diastolic arterial BP were measured using a non-invasive automatic blood-pressure monitor (Boso Medicus Juningen, Germany) at time points shown in Figure 1.

### 2.8. Alpha-Amylase and Cortisol

Saliva samples were collected at 5 time points during the experimental session and once on the training day (Figure 1). Following established procedures [42], participants were instructed to chew lightly on a Salivette^®^ cotton swab (Salivette bomull, Sarstedt, Landskrona, Sweden) for 60 s and to move around the swab while chewing. After each study session, cotton swabs were frozen at −18 °C. All saliva samples were analyzed for sAA. Three samples were analyzed for cortisol: (1) the baseline sample at the beginning of each session, (2) the sample between work-bouts 6 and 7, and (3) the sample from the end of each session. Sample 2 was collected between bouts 6 and 7 instead of mid-way in the work bout in order to secure a sufficient amount of saliva.

sAA activity was determined using the method of Pointe Scientific, Inc. (Liquid Amylase, CNPG3 reagent set) and expressed as units per ml saliva (U/mL saliva). Thawed saliva samples were centrifuged at room temperature for 2 min at 1000 g. The supernatant was diluted 1:50 in distilled water and a ten microliters sample was used for duplicate analyses. Enzyme activity was measured by the absorbance increase in the assay medium at 405 nm for 3 min at 37 °C. Blank reaction was assessed by measuring activity in distilled water (10 µL). All saliva samples were analyzed for sAA.

For cortisol analyses, duplicate determinations were made in 40 µL of centrifuged saliva from the sample used to determine sAA. Free salivary cortisol was analyzed using an enzyme immunoassay kit (Cortisol Saliva ELISA, SE 120038; Sigma-Aldrich, Darmstadt, Germany) and expressed as ng cortisol/mL saliva.

### 2.9. Statistical Methods

When planning the study, we performed power calculations, showing that a minimum of 15 participants were needed to sufficiently reduce the likelihood of type II error in repeated-measures studies of the addressed outcomes, given what was considered relevant effect sizes.

Assumptions of normally distributed data were checked by calculating skewness and kurtosis. Non-normal distributions were considered to be present if these parameters exceeded +2 or −2.

#### 2.9.1. Baseline Observations

For descriptive purposes, variables were expressed as group means with standard deviations (SD) between participants. Differences between protocols in subjective (ratings) and objective (HR, RMSSD, HF, cortisol, sAA) indicators before the experiment, i.e., pre-test, were examined using repeated-measures ANOVA with CT1-3 as a within-subjects factor.

#### 2.9.2. Effects of Alternations

Effects of the experimental work bouts on perceived stress during the last minute of each pipetting work bout and after each CT work bout were tested using a set of repeated-measures ANOVAs, with CT (three levels), time (work bouts; ten levels), and their interaction as within-subject factors.

To determine the effect of alternating physical and cognitive work on physiological stress indicators, a set of repeated-measures ANOVAs (effects: CT, time, and interaction CT × time) were run with data expressed in percentage of baseline values. Time effects were addressed by examining ten observations of HR and HRV-variables during work (i.e., the mean value in each work-bout); two levels for systolic BP, diastolic BP, and cortisol (i.e., mean values of pre- and post-measurements); and four levels for sAA.

In all ANOVAs, the order of experiments differing in CT difficulty was added as a between-subject factor to account for possible order effects. In all statistical tests, partial eta squared (ηp^2^) was used to express effect size, tentatively classified according to Cohen (1988) [65], i.e., small, medium and large effects corresponding to ηp^2^ of 0.01, 0.06, and 0.14, respectively.

All statistical analyses were performed in SPSS version 24.0 (IBM SPSS Statistics for Windows, IBM Corp, Armonk, NY, USA).

## 3. Results

### 3.1. Data Inspection

We found no critical violations of assumptions of normally distributed data, except for HF HRV, which were therefore log-transformed prior to further analysis.

### 3.2. Baseline Value

At baseline, i.e., before performing any physical or cognitive task, we found no significant differences between the three conditions (CT1-3) in any outcome variable (all *p* > 0.07, all ηp^2^ < 0.17), except for BP. Systolic BP was slightly higher before experiments with CT3 compared to CT1 (*p* = 0.04, ηp^2^ = 0.27), while diastolic BP was slightly higher before CT1 compared to CT2 (*p* = 0.02, ηp^2^ = 0.34).

### 3.3. Perceived Stress

Stress ratings during the last minute of each pipetting work bout were not significantly associated with CT difficulty (Table 1) and did not change significantly across time. Stress ratings after each CT-bout increased significantly across time (Table 1, Figure 3), but the time effect did not differ significantly between CT difficulty levels (interaction CT × time).

### 3.4. Physiological Indicators of Stress

#### 3.4.1. Heart Rate and HRV

Data on HRV from one participant were excluded due to technical problems. HR, RMSSD and HF changed significantly across consecutive work bouts (Table 2); HR decreased across time (Figure 4), while RMSSD and HF increased (Figure 5 and Figure 6), suggesting an overall increased parasympathetic activity. CT difficulty did not affect HR, RMSSD, or HF significantly, and we found no significant interaction between CT and time.

#### 3.4.2. Blood Pressure

Systolic and diastolic BP did not change significantly between pre- and post- test batteries, and they were not significantly associated with CT difficulty. We found no significant CT × time interaction effects (Table 2).

#### 3.4.3. Salivary Alpha Amylase (sAA) and Cortisol

Experimental session time (morning, afternoon or mixed) influenced sAA activity (F = 5.69, *p* = 0.03, ηp^2^ = 0.56), but not cortisol (F = 0.24, *p* = 0.79, ηp^2^ = 0.05).

sAA activity was higher during CT1 than during CT2 and CT3 (Figure 7), and this difference was statistically significant (Table 2, Figure 7). Thus, sympathetic nervous activity was larger during the easy CT than during the moderate and difficult CTs. CT difficulty had no main effect on cortisol, and we found no significant CT × time interaction.

sAA did not change significantly with time (Table 2). However, in the CT2 and CT3 experiments, sAA baseline values were higher than immediately before the subsequent pipetting work. Cortisol decreased with time, but the time effect was not significant (Table 2).

## 4. Discussion

We examined the effects on stress-related responses of 110 min alternating physical and cognitive work, the latter at three different difficulty levels. Contrary to our hypotheses, we found no significant increase in perceived stress, BP, salivary cortisol, or sAA during the work bout, irrespective of the cognitive task difficulty. Rather, HR decreased, and HRV increased significantly over time, indicating increasing parasympathetic activation. We found no effect of CT difficulty on stress-related responses, apart from sAA increasing more from baseline levels during work including the easiest cognitive tasks (CT1) compared with the moderate (CT2) and difficult (CT3) tasks. Overall, our findings suggest that work consisting of a light, repetitive physical task alternating with a cognitive task, at the intensity levels and temporal patterns investigated here, is not associated with any marked stress response, regardless of cognitive task difficulty.

### 4.1. Stress Effects of Combining Physical and Mental Work

To our best knowledge, this is the first study to investigate alternating physical and cognitive tasks using a comprehensive selection of stress indicators, including both subjective (perceived stress) and objective indicators, with the latter covering both ANS branches, i.e., sympathetic nervous activation (amylase), vagal modulation (HRV-indices), and the HPA-axis (cortisol). We expected that 110 min of work would result in a stress response, and that this response would be more pronounced for a more difficult cognitive load, but no such effect was observed. The CT could be performed throughout the 110-min working period without any notable performance reduction [38] despite CT performance differing significantly between the three CT difficulties (correct positive answers *p* < 0.001, ηp^2^ = 0.70; false positive answers *p* < 0.001, ηp^2^ = 0.49 [38]), which could suggest differences in stress. Moreover, perceived mental effort differed between CT levels (*p* = 0.01, ηp^2^ = 0.30), even though mental fatigue ratings did not (*p* = 0.29, ηp^2^ = 0.12) [38].

Mathiassen et al. [39] examined cardiovascular stress markers (HR, HRV, and BP) during alternations between repetitive physical and cognitive tasks and found that alternations led to elevated levels of HR and BP that persisted above baseline-levels one hour after work. In contrast, we found that HR decreased, and HRV increased (rMSSD and HF power), indicating that parasympathetic nervous activity increased across the ten work bouts. One explanation for these unexpected findings may involve the fairly low intensity of the physical task (i.e., pipetting). Previous studies have found that sustained isometric contractions at intensities exceeding 10% MVC lead to increased sympathetic nervous activity [66] and that concurrent mental demands have more pronounced effects on parasympathetic withdrawal at higher intensities of physical work [67].

The stress response is also likely to depend on the cognitive task. The n-back task used in this study did not expose participants to multiple stressors, while this has often been the case in previous experiments of mental tasks alone or concurrent with a physical task. For instance, Hjortskov et al. [30] studied effects of mental stress during computer work on HRV-indices and BP, using a CT combined with socially threatening exposures, i.e., aggression and surveillance. The stress condition led to a vagal withdrawal, although BP was higher during computer work alone than when adding stress. Krantz et al. [32] investigated physiological stress responses to three versions of the Stroop Color Word Test and two types of arithmetic tasks. All stressors increased levels of sympathetic activity (catecholamines, HR, systolic and diastolic BP), while—consistent with our results—none of these protocols significantly increased HPA-axis activity. In contrast to our study, where conditions were performed at separate days, all mental stressors in Hjortskov et al. [30] and Krantz et al. [32] were performed within the same experimental session, and carry-over effects may have occurred.

### 4.2. Ecological Validity

Previous experimental studies of concurrent and alternating physical and cognitive tasks include examples of lighter physical tasks than pipetting, such as computer work [30], but also considerably heavier tasks, such as maximal isokinetic leg extensions or manually moving a 300 g manipulandum [39]. We argue that the pipetting task in our study has a good ecological validity and can serve as a valid model of low-intensity repetitive upper extremity tasks commonly performed in occupational settings [52,53,54]. The pipetting sequence used in the present study, i.e., switching sequentially between tubes in a predictable pattern, was designed to impose only a minor cognitive load, mimicking the situation in simple repetitive occupational tasks. Pipetting task performance remained stable throughout the work bouts with very few errors occurring during any of the experimental sessions; this agrees with previous findings regarding this pipetting task [68]. While frequent shifts between tasks has been suggested to lead to reduced productivity due to task interruptions [69], we observed no such effects. We emphasize that the choice of pipetting, i.e., a low-intensity physical task, likely resulted in less pronounced stress responses than those following from performing more intense or heavy physical tasks. However, our aim in the present study was not to deliberately provoke any stress response but to investigate the extent to which a stress response would develop during a typically occurring task.

In selecting the n-back task, we prioritized having a CT that was easily manipulated and controlled, rather than adopting a task with a closer resemblance of occupational work, such as having demanding instructions for how to perform the pipetting task correctly [49]. Thus, the cognitive demands associated with n-back are likely less complex than those occurring in many occupational tasks with multiple parallel stressors. Still, we find that n-back is occupationally relevant, in involving a number of key working memory processes also occurring in common work tasks, such as processing information, keeping it in the memory, and updating it. Thus, n-back reflects occupational demands better than, e.g., simple span tasks, reaction time tests, or the Stroop color word test [56]. N-back also activates brain areas involved in error detection and strategic reorganization of information [57], which are processes involved in everyday work.

The temporal pattern of alternations between physical and mental tasks is likely to influence the physiologic response [6,7], but specific associations remain to be investigated. Some studies of rest breaks suggest that shorter and more frequent breaks are more effective in reducing fatigue and muscle discomfort than longer and less frequent breaks [69,70,71,72], but in general, research addressing temporal patterns appears inconclusive [9]. Future studies should investigate to which extent different temporal alternation patterns influence stress responses.

Our experimental design with a strictly controlled temporal pattern only serves as a crude model of an occupational setting. Worker autonomy will probably be greater in real-life occupational settings, leading to more irregular alternation patterns. Moreover, additional conditions present at work but not in the laboratory, e.g., external demands and support from colleagues, will likely influence stress responses [73,74], and maybe interact with the demands in cognitive tasks. In addition, stress responses may develop if work continues for more than 110 min [75], and we encourage studies extending the total exposure duration.

Notably, we did not attempt to identify a “just right” combination of productive physical and cognitive task(s) that would be sufficiently challenging to promote both mental and physical performance and health, despite this being an ultimate goal in job design according to the Goldilocks work paradigm [76]. Pipetting is unlikely to provide sufficient stimulus to improve cardiorespiratory or musculoskeletal capacity. Linking to the Goldilocks work paradigm, future controlled studies of alternating physical and mental work tasks should consider using physical work tasks with different, and maybe larger, load in time patterns inspired by, e.g., exercise and sports physiology.

### 4.3. Methodological Considerations

A strength of our study is the comprehensive approach in monitoring stress responses, including both subjective (perceived stress) and objective indicators, representing both ANS branches, and the HPA-axis.

However, the technique used for saliva samples may have disadvantages. The Salivette^®^ is easy to administer, but since it is absorbent, saliva flow can be compromised. Moreover, the collection method does not account for saliva dilution [77]. This is important when interpreting the sAA and cortisol values. Thus, future studies should consider using methods accounting for saliva flow.

Our results are based on a limited number of participants (n = 15), which may, for some of the outcomes, lead to a critical risk of type II errors. In addition, the small sample size in combination with the convenience recruitment of participants may add uncertainty regarding the representativeness of our sample vis-a-vis the target population, i.e., healthy women, 20 to 50 years of age, with previous pipetting experience.

Women are more often than men engaged in repetitive physical work tasks [78] and also report more MSD [45]. However, the choice of women as a study group requires caution in generalizing results to men, since men may respond differently to both the physical and the cognitive task. Our sample consisted of quite young women, and the results may not transfer to older individuals, considering that both cognitive and physical performance change with age [79,80]. In addition, individuals with stress-related health problems may experience alternating tasks as more stressful than individuals with no such problems [25,81]. Moreover, individual differences in executive functioning seem to influence stress regulation [82]. Thus, effects of CT difficulty on stress may vary between individuals having different working memory capacities.

## 5. Conclusions

Alternations between a light, repetitive physical work task and a cognitive task did not lead to any pronounced stress responses (perceived stress and indicators of sympathetic and parasympathetic nervous activity) among healthy women. Cognitive task difficulty did not influence the stress response, except for a greater increase in sAA-levels from baseline during the easiest CT. 

Thus, our results suggest that combining physical and cognitive tasks may be an option in job rotation and that cognitive task difficulty seems to be less of a concern, at least when demands are within the limits investigated here.

## Figures and Tables

**Figure 1 ijerph-17-08509-f001:**
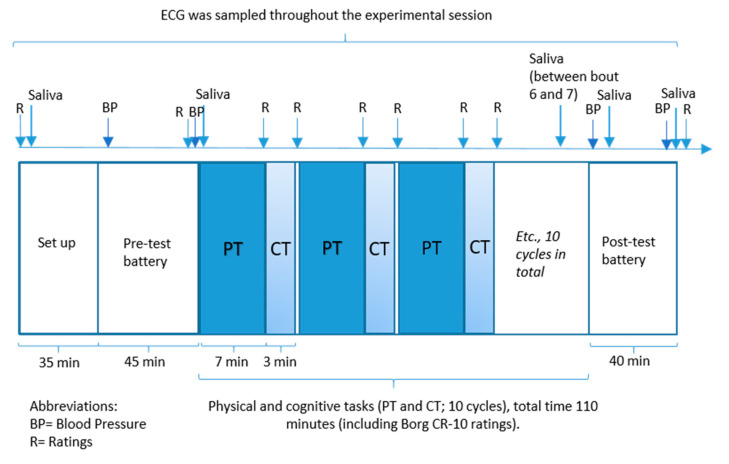
Flow chart for the experimental session. Total session duration was approximately four hours. ECG: Electrocardiography.

**Figure 2 ijerph-17-08509-f002:**
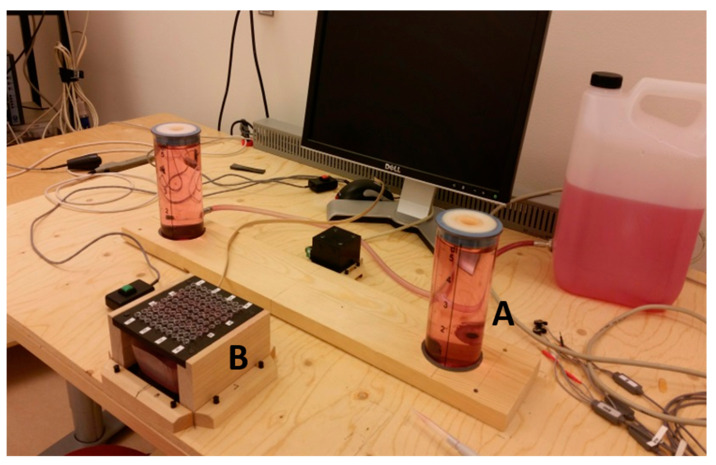
Arrangement of target and pickup tubes; the right, larger tube (A) was the pickup tube, and liquid was to be dispensed into small tubes in the fixture at the front (B).

**Figure 3 ijerph-17-08509-f003:**
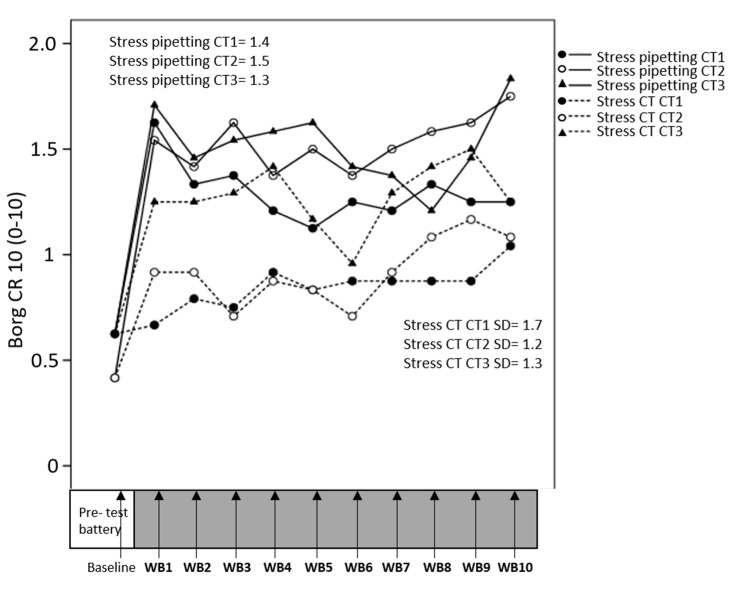
Development of perceived stress across consecutive work bouts (WB) of pipetting and cognitive task (CT). Each time point on the x-axis represents ratings of stress during the last minute of pipetting (full drawn lines) and just after each CT bout (dotted lines). Standard deviations for means across all participants and all time points.

**Figure 4 ijerph-17-08509-f004:**
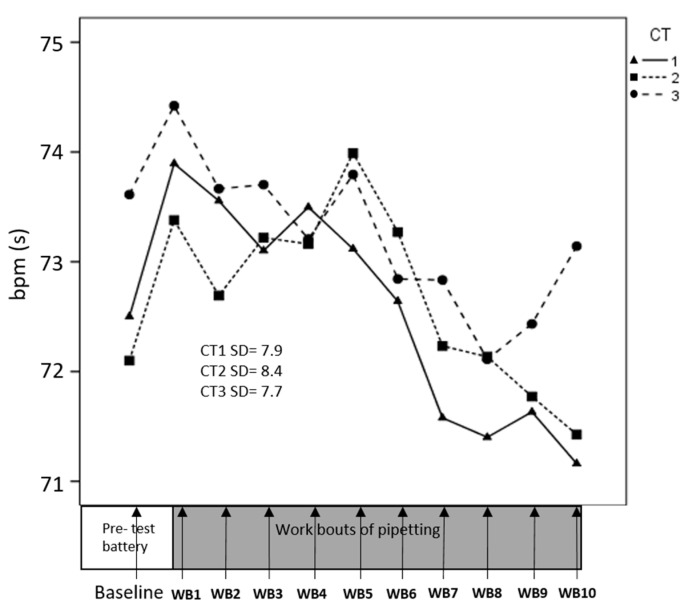
Heart rate (HR) across consecutive pipetting work bouts (WB). At each time point on the x-axis, corresponding to the pipetting work bouts, a mean value for HR across participants is presented. Separate lines illustrate the three cognitive task (CT) difficulties. Standard deviations (SD) refer to all participants and all time points.

**Figure 5 ijerph-17-08509-f005:**
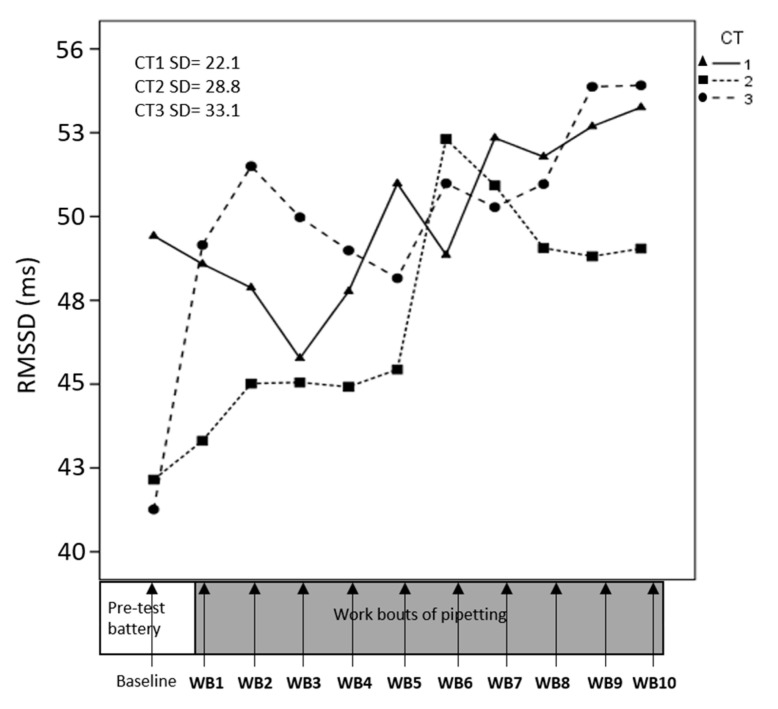
Development of root mean square of the successive differences between R–R intervals (RMSSD) across consecutive pipetting work-bouts (WB). At each time point on the x-axis, corresponding to the pipetting work bouts, a mean value for RMSSD across participants is presented. Separate lines illustrate cognitive task (CT) difficulties. Standard deviations for means across all participants and all time points.

**Figure 6 ijerph-17-08509-f006:**
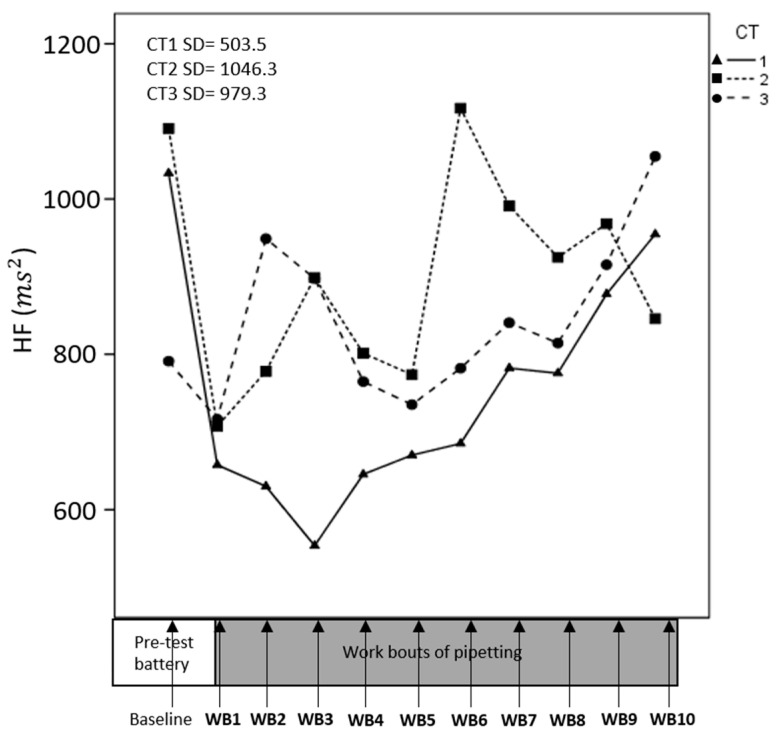
Development of High Frequency spectral power (HF) across consecutive pipetting work bouts (WB). At each time point on the x-axis, corresponding to the pipetting work bouts, a mean value for HF across participants is presented. Separate lines illustrate cognitive task (CT) difficulties. Standard deviations for means across all participants and all time points.

**Figure 7 ijerph-17-08509-f007:**
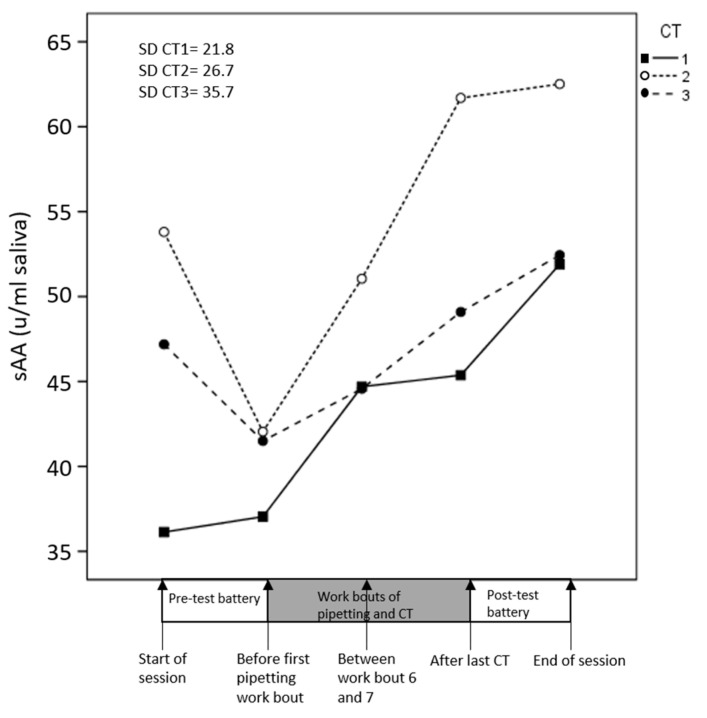
Salivary alpha-amylase (sAA) during the experimental session. Separate lines illustrate the three cognitive task (CT) difficulties. Standard deviations (SD) refer to all participants and all time points.

**Table 1 ijerph-17-08509-t001:** Ratings of stress during the last minute of pipetting bouts and just after cognitive task (CT) bouts. Results of repeated measures ANOVAs with main effect and interactions, and mean values (with SD) in each CT condition across all points in time.

Variable	ANOVA				Mean (SD)		
	df	F	*p*	ηp^2^	CT1	CT2	CT3
**Stress (CR-10)** **during pipetting**							
Time	9	0.74	0.67	0.08			
CT	2	1.36	0.28	0.13	1.7 (1.4)	1.9 (1.5)	1.8 (1.3)
Interaction (CT × time)	18	0.59	0.90	0.06			
**Stress (CR-10)** **after CT**							
Time	9	2.19	0.03	0.15			
CT	2	2.74	0.09	0.19	1.2 (1.7)	1.3 (1.2)	1.5 (1.3)
Interaction (CT × time)	18	0.78	0.73	0.06			

Abbreviations: df = degrees of freedom; ηp^2^ = partial eta squared; CT = cognitive task; SD = standard deviation between participants; CR = category ratio.

**Table 2 ijerph-17-08509-t002:** Physiological indicators of stress (heart rate, heart rate variability (HRV), blood pressure (BP), salivary alpha amylase (sAA)), and cortisol during work. Results of repeated measures ANOVAs with main effect and interactions, and mean values in each CT condition across all points in time.

Measure	ANOVA				Mean (SD)		
	Df	F	*p*	ηp^2^	CT1	CT2	CT3
**HR (bpm)**							
Time	9	9.37	≤0.001	0.33			
CT	2	0.76	0.48	0.06	72.6 (8.0)	72.7 (8.4)	73.2 (7.7)
Interaction (CT × time)	18	1.23	0.24	0.08			
HRV							
**RMSSD (ms)**							
Time	9	2.51	0.01	0.16			
CT	2	0.29	0.75	0.02	50.0 (22.1)	47.4 (28.8)	50.8 (33.1)
Interaction (CT × time)	18	0.40	0.99	0.03			
**HF (log ms^2^) ***							
Time	9	4.17	≤0.001	0.26			
CT	2	0.32	0.72	0.03	2.7 (0.3)	2.7 (0.4)	2.7 (0.3)
Interaction (CT × time)	18	0.74	0.77	0.06			
**BP (mmHg)**							
*Systolic*							
Time	2	1.91	0.29	0.28			
CT	2	1.76	0.22	0.26	112.9 (7.8)	112.6 (8.9)	113.8 (7.9)
Interaction (CT × time)	4	1.18	0.35	0.19			
**Diastolic**							
Time	2	0.80	0.48	0.14			
CT	2	0.88	0.44	0.15	73.6 (7.5)	72.4 (7.0)	73.3 (6.0)
Interaction (CT × time)	4	1.95	0.14	0.28			
**sAA (µ/mL saliva)**							
Time	3	1.38	0.27	0.13			
CT	2	4.86	0.02	0.34	43.0 (21.8)	49.6 (26.7)	54.2 (35.7)
Interaction (CT × time)	6	0.28	0.94	0.03			
**Cortisol (ng cortisol/mL saliva)**							
Time	1	0.32	0.59	0.03			
CT	2	0.09	0.92	0.01	8.2 (2.7)	7.9 (2.0)	8.0 (3.1)
Interaction (CT × time)	2	1.97	0.17	0.18			

Abbreviations: df = degrees of freedom; ηp^2^ = partial eta squared; CT = cognitive task; SD = standard deviation between participants. * HF was log-transformed prior to further analysis. Data were expressed in percentage of baseline values for all outcomes.

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
