# Peer review of "Stress-Related Responses to Alternations between Repetitive Physical Work and Cognitive Tasks of Different Difficulties"

_ijerph, 2020, doi:10.3390/ijerph17228509_

Round 1

Reviewer 1 Report

Introduction:
In general, the wording does not detail specific and concrete data from the studies cited, but rather global aspects are indicated.
Lines:

35: increasing rest, ¿how many? 

39-40: what is the limited duration?

56: Do only stress experiences outside of work influence?

61: In which studies?

69-70: Could you provide any study that does not correspond to a self-citation?

77-78: Can you give some specific data about MSD in women?

Materials and Methods

Can other data be provided indicating that a sample of 15 women may be representative for obtaining conclusive results on the selected population profile?

Lines:

135: Is the term letter correct or is it better to use the term word?

172: Perceived stress has been measured with the Borg CR-10 scale, but it is indicated that this scale is mainly used to evaluate physical effort and fatigue. Therefore, can they be considered a reliable instrument to assess perceived stress?

213-216: Confusing paragraph, clarify in its wording, please

Results

In general, in each section of the results, the data obtained are shown, but it is not explained what an increase or decrease in each of them means. It should be detailed to facilitate understanding of the conclusions reached later.

235-237: Can you provide a table with the indicated data?

249-251: Make the title shorter, giving only a title for the table and include in the previous paragraph the most developed interpretations of the table

253-257: Make the title shorter, putting only one title for the figure and detailing it better in the next paragraph, as well as the legend of the figure

268-271: Comment like Table 1

274-288: Comment like Figure 3

Discussion

The discussion includes aspects that belong to the conclusions section.

It is recommended to include in the discussion those aspects of the study that differ from or are reinforced by other studies by making the corresponding bibliographic citations, as included in the article.

It is recommended to include in the conclusions those aspects that are not compared with other studies and are obtained only from the applied result of the research carried out.

Lines:

326-337-341-345: different format of bibliographic citation

330-331: Can the physical task selected in the research be considered too low to obtain results that can be extrapolated to the workplace?

335-337: The n-back task does not include multiple stressors, does this mean that the initial hypothesis in the research could not be extrapolated to a real situation in the workplace about the cognitive task?

349: mistake in the citation (cf.28) with different format

351-353: It has been indicated before that it could be too light a physical task, could it be explained so that the reader does not consider it contradictory?

360-365: This paragraph may be interpreted in a manner contrary to lines 335-337 Could it be clarified in its wording?

375-381: It is therefore indicated that the initial hypotheses cannot be considered conclusive since they could be affected by other parameters that occur in a working environment and have not been included in the research.

Consequently, how could these results be considered relevant? could you include or clarify them?

Conclusions:

Expand with aspects that have been included in the discussion, as indicated above

Author Response

Overall response for reviewer 1:

We sincerely appreciate the time you have spent in reviewing our manuscript and want to thank you for your important and relevant comments. We have addressed all of them below point by point, and revised the manuscript accordingly.

Comments and Suggestions for Authors

Introduction:
1. In general, the wording does not detail specific and concrete data from the studies cited, but rather global aspects are indicated.

Reply: We thank you for this general comment about the introduction section. We have addressed specific comments regarding the introduction section below, and we hope that these answers will be satisfactory. Also, we prefer the introduction to not be too lengthy, and to save more detailed aspects of previous research to the discussion section, where we can compare them to our own findings.  

Lines:

  1. 35: increasing rest, ¿how many? 

Reply: In this paragraph, at the very beginning of the introduction, we want to provide the reader with a general/’generic’ basis for to how to influence exposure variation, since this is relevant to the eventual aim of the present study. One way of manipulating variation in biomechanical exposure is to change the duration of rest allowances; in most cases for the purpose of offering recovery from fatiguing physical work. While we refer to controlled studies addressing this issue (Mathiassen 1993; Konz 1998; Dickerson et al. 2015), we cannot give any explicit answer to ‘how many’ rest periods would be needed to increase variation; simply because there is no definitive answer to this, and – as said above - this is a general example on how to influence exposure variation. We have, however, edited the text to better reflect our general view on the generic effect of breaks on variation (line 35-36, revised manuscript).

  1. 39-40: what is the limited duration?

Reply: We thank you for this comment. We intend this to be a general, ‘generic’ statement about rest breaks in production, and we do not believe that there would exist any ‘universal’ answer to how much ‘unproductive’ time would be allowed in a production; this will depend profoundly on the trade and specific company. We have, however, rephrased the sentence to avoid confusion regarding the term “limited duration” (lines 39-40 revised manuscript).

  1. 56 (lines 55-56 revised manuscript): Do only stress experiences outside of work influence?

Reply: We realize that the explanation was unclear and that we need to emphasize the role of psychosocial demands in influencing the risk of developing MSDs. We have edited this text accordingly (lines 55-56 in the revised manuscript).

  1. 61: In which studies?

Reply: We have added references to the statement “However, sufficient recovery…promote health” (line 61 in revised manuscript).

  1. 69-70: Could you provide any study that does not correspond to a self-citation?

Reply: As part of our research activities investigating occupationally relevant alternations of physical and mental tasks, we have conducted an extensive systematic literature search for original studies addressing this issue. To the best of our knowledge, no other research group has conducted any controlled studies on the effect of alternating physical and cognitive tasks in an occupational context.

  1. 77-78 (lines 78-80 revised manuscript): Can you give some specific data about MSD in women?

Reply: We have added a reference supporting this statement in the manuscript (lines 78-80 revised manuscript). We emphasize that the literature on MSD differences between men and women is very rich, and that numeric differences in prevalence or incidence depend profoundly on the specific MSD (shoulder, low-back, …), the criterion for MSD (self-report, medical examination, …), and the addressed trade, country, and population.  

Materials and Methods

  1. Can other data be provided indicating that a sample of 15 women may be representative for obtaining conclusive results on the selected population profile?

Reply: Our sample size is, indeed, small and we agree that 15 women may not necessarily represent the entire target population. When planning the study, we performed power calculations, showing that a minimum of 15 participants was needed to arrive at an acceptable power in a study with the expected/relevant differences between protocols in a typical outcome among those we assessed. The design of the study (controlled experiment with a within-person repeated-measures design) was chosen to strengthen power, since repeated-measures designs have stronger statistical properties than between-subject designs. We have revised the discussion section to elaborate on the issue of representativeness (lines 427-430 and 433-435), emphasizing that the results from our study may not be readily transferable to other study populations, such as older people and men, and that the effects will vary depending on individual differences in executive functions (see revised manuscript lines 437-439). We have revised the method section to incorporate a text about power analysis (revised manuscript lines 221-223).

Lines:

  1. 135 (line 140 revised manuscript): Is the term letter correct or is it better to use the term word?

Reply: The term letter is correct since our n-back test presented the participants with a single letter (one of seven consonants; r, m, w, s, q, k, t) and not a word.

  1. 172 (lines 181-182 revised manuscript): Perceived stress has been measured with the Borg CR-10 scale, but it is indicated that this scale is mainly used to evaluate physical effort and Therefore, can they be considered a reliable instrument to assess perceived stress?

Reply: We agree that the wording in the manuscript is confusing, as it indicates that the Borg CR-10 scale may only be used to measure physical effort and fatigue. We have revised the manuscript to emphasize that the CR-10 scale is a validated, general intensity scale which has been used before to measure perceived stress (E. Borg 2007; Borg 1998). We have explained this in the manuscript (lines 181-182 revised document).

  1. 213-216 (line 228-229 revised manuscript: Confusing paragraph, clarify in its wording, please

Reply: We have revised the paragraph, by rephrasing the text, and hope that this has increased clarity (see manuscript, lines 228-229).  

Results

  1. In general, in each section of the results, the data obtained are shown, but it is not explained what an increase or decrease in each of them means. It should be detailed to facilitate understanding of the conclusions reached later.

Reply: We have revised the results section to provide more details, as suggested by this comment. (lines 276-277, 316-317).

  1. 235-237 (224-225 revised manuscript): Can you provide a table with the indicated data?

Reply: We appreciate the comment, but we would prefer not to include a lengthy table with values of skewness and kurtosis for all variables in all protocols. However, we have clarified that normality was assessed through calculations of skewness and kurtosis, with values exceeding +2 or -2 disqualifying a normal distribution (lines 224-225 revised manuscript).   

  1. 249-251 (263-265 revised manuscript): Make the title shorter, giving only a title for the table and include in the previous paragraph the most developed interpretations of the table

Reply: We have revised and shortened the title of the table (lines 263-265 revised manuscript).

  1. 253-257 (268-271 revised manuscript): Make the title shorter, putting only one title for the figure and detailing it better in the next paragraph, as well as the legend of the figure

Reply: While we appreciate the comment, we believe that the information provided in the legends to the figures is needed to help the reader to understand how to properly read the figure without having to look for additional help in the running text; this is generally recommended by most scientific journals. We have, therefore, decided not to revise the figure legends.

  1. 268-271 (289-291 revised manuscript): Comment like Table 1

Reply: We have revised and shortened the title of the table (line 289-291 revised manuscript).

  1. 274-288 (296-299 revised manuscript): Comment like Figure 3

Reply: We refer to our answer to question 15.

Discussion

  1. The discussion includes aspects that belong to the conclusions section. It is recommended to include in the discussion those aspects of the study that differ from or are reinforced by other studies by making the corresponding bibliographic citations, as included in the article.

Reply: It is recommended to include in the conclusions those aspects that are not compared with other studies and are obtained only from the applied result of the research carried out.

We thank you for these general comments about the discussion section. We have made general changes accordingly and we hope that these will be satisfactory.

Lines:

  1. 326-337-341-345 (line 348-368 in revised manuscript): different format of bibliographic citation

Reply: We have adjusted the citations (see manuscript, lines 348-368).

  1. 330-331 (lines 352-356 revised manuscript): Can the physical task selected in the research be considered too low to obtain results that can be extrapolated to the workplace?

Reply: The main focus intention of this study was not to provoke any pronounced response in the stress regulatory systems by designing a study with “extreme” physical and cognitive loads, but rather to investigate to what extent combinations of normally occurring work tasks result in stress responses. We argue (lines 374-376) that the pipetting task is occupationally relevant. Pipetting is performed in many occupations, and may serve as a good model of low-intensive, repetitive task engaging the upper extremity (Björksten, Almby, and Jansson 1994; Fredriksson 1995; Lintula and Nevala 2006). We have clarified and expanded this view in the revised manuscript (lines 382-386).

  1. 335-337 (357-359 revised manuscript): The n-back task does not include multiple stressors, does this mean that the initial hypothesis in the research could not be extrapolated to a real situation in the workplace about the cognitive task?

Reply: Indeed, the n-back task does not include multiple stressors. The contrasting results in our study are likely due to the use of a productive cognitive task, which was not intended to induce stress. This should be seen in contrast to the studies described in the discussion section, with the intention to induce stress by incorporating classical stress tasks (e.g. threating situations or conflicting information. We have also added this in the manuscript (line 389-391 revised manuscript).

  1. 349 (line 372 revised manuscript): mistake in the citation (cf.28) with different format

Reply: We have corrected this formatting error (see manuscript, line 372).

  1. 351-353 (line 352-354 revised manuscript): It has been indicated before that it could be too light a physical task, could it be explained so that the reader does not consider it contradictory?

Reply: As explained in our answers to question 20, we have revised the section on ecological validity to clarify our ideas and viewpoints.

  1. 360-365 (line: 387-393 revised document). This paragraph may be interpreted in a manner contrary to lines 335-337 (line 357-359 revised manuscript). Could it be clarified in its wording?

Reply: Please see answer to question 22.

  1. 375-381 (line 397-402 revised manuscript): It is therefore indicated that the initial hypotheses cannot be considered conclusive since they could be affected by other parameters that occur in a working environment and have not been included in the research.

Reply: We agree that the sentence is misleading. We intended to say that our study protocol did not capture different temporal patterns of alternations which needs to be addressed in further research. We would also like to emphasize that our study design does not address all relevant parameters in the working environment. We have emphasized this in the revised manuscript, in particular in the sections on ecologic validity and study limitations (revised manuscript, line 403-404).

  1. Consequently, how could these results be considered relevant? could you include or clarify them?

Reply: As argued above, and in the sections on ecological validity and study limitations, we believe that our results – with the caveats now emphasized and discussed – are relevant to occupational practice, and as an input to the discussion of how to construct effective job rotations in occupational life.

Conclusions:

  1. Expand with aspects that have been included in the discussion, as indicated above

Reply: We have revised the conclusion section and included some of the aspects which was earlier embedded in the discussion section (line 442-452 revised document).

Reviewer 2 Report

The study addressed the problem of the possible mixing of physical and cognitive tasks without causing excessive stress, with several objective measures and a subjective Borg CR-10 scale, on healthy women.

The results indicated that cognitive task difficulty appeared not to have a significant increase in stress, although difficult cognitive task caused a significant increase in stress by itself.

The results are interesting and deserve to be published, even with the possible drawbacks that the intensity of the physical task may not be heavy enough and that hard enough cognitive work has not been tried.

Author Response

Overall response for reviewer 2:

Thank you for taking the time to review our manuscript. We have responded below.

  1. The study addressed the problem of the possible mixing of physical and cognitive tasks without causing excessive stress, with several objective measures and a subjective Borg CR-10 scale, on healthy women.

Reply: The results indicated that cognitive task difficulty appeared not to have a significant increase in stress, although difficult cognitive task caused a significant increase in stress by itself.

The results are interesting and deserve to be published, even with the possible drawbacks that the intensity of the physical task may not be heavy enough and that hard enough cognitive work has not been tried.

The aim of the study was to investigate to what extent alternations between a light, repetitive physical task and a cognitive task have an impact on stress responses. The tasks were selected to represent demands occurring in occupational work, i.e. to be occupationally relevant, and not to explicitly result in a pronounced stress response.

Reviewer 3 Report

This manuscript is a well designed study that aimed to determine the extent to which stress develops during alternating physical and cognitive work in a group of women. The only question that I have is about the procedure. What are WB1, WB2, and etc. in Figures 3-7? I supposed they are different time points of measurement. I don't understand why there are six lines if all participants perform cognitive tasks in a random order of difficulty levels during a single trial.

Author Response

Overall response for reviewer 3:

We thank you for accepting to review our manuscript. We have done our best to answer your question below, and hope that our answer will be satisfactory. 

  1. This manuscript is a well designed study that aimed to determine the extent to which stress develops during alternating physical and cognitive work in a group of women. The only question that I have is about the procedure. What are WB1, WB2, and etc. in Figures 3-7? I supposed they are different time points of measurement. I don't understand why there are six lines if all participants perform cognitive tasks in a random order of difficulty levels during a single trial.

Reply: We have tried to address the comment according our interpretation. It is correct that WB1, WB2 etc on the x-axis represent time points for measurements/data collection. Each separate line illustrates one of the three study conditions. Each study condition addressed alternations between pipetting and the cognitive n-back task in only one of three difficulty levels (easy, moderate or difficult); thus, each of these n-back conditions were performed on separate days (not all three during a single trial). In figure 3, however, (which we believe you refer to), we have six lines instead of three. The full drawn lines represent stress ratings during the last minute of pipetting work bouts and the dotted lines illustrates stress ratings just after each CT work bout.

Reviewer 4 Report

This study investigated stress-related responses during alternating physical and cognitive work. The results show no significant alternations between physical and cognitive tasks. Overall, I think it is a well-written manuscript. Especially, dependent and independent variables are clearly introduced in the Method section. Having had the opportunity to review this manuscript, I like to provide the following suggestions to improve the quality of the manuscript.

First, the environment that subjects perform physical and cognitive tasks should be addressed, such as temporal, humidity, noise, luminance, etc.

Second, the authors used repeated-measures ANOVA to compare changes in measured variables. However, there are only 15 subjects. How the sample size was derived and whether repeated-measures ANOVA is appropriate should be justified.

Third, p.5, line 162, section 2.6 Pre- and post-test battery: It’s unclear to me why the experiment proceeded only at negative answers to the two latter questions. Should sleep quality, 24-hour physical activity, and consumption of drinks be considered?

Finally, the authors introduced some abbreviations. They should be introduced for the first time they occur and should not be introduced redundantly. For example, "musculoskeletal disorders (MSD)" appears twice on line 34 and line 54.

Author Response

Overall response for reviewer 4:

Reply: Thank you for taking your time to review our manuscript, and for the valuable comments. We have read and addressed each of them below to the best of our ability, and we hope that the response and related revisions in the manuscript will be satisfactory.

Comments and Suggestions for Authors

This study investigated stress-related responses during alternating physical and cognitive work. The results show no significant alternations between physical and cognitive tasks. Overall, I think it is a well-written manuscript. Especially, dependent and independent variables are clearly introduced in the Method section. Having had the opportunity to review this manuscript, I like to provide the following suggestions to improve the quality of the manuscript.

  1. First, the environment that subjects perform physical and cognitive tasks should be addressed, such as temporal, humidity, noise, luminance, etc.

Reply: We have added details about the environmental factors in the methods section, to the extent possible (see revised manuscript, lines 102-105).

  1. Second, the authors used repeated-measures ANOVA to compare changes in measured variables. However, there are only 15 subjects. How the sample size was derived and whether repeated-measures ANOVA is appropriate should be justified.

Reply: In planning the study, we performed power calculations, showing that a minimum of 15 participants were needed to sufficiently reduce the likelihood of type II error in repeated-measures studies of the addressed outcomes, given what was considered relevant effect sizes. RM-ANOVA is a robust statistical test for repeated measurements, because it controls for factors that cause variability between subjects; thus, within-subject repeated-measures designs have greater statistical power than for instance between-group designs (Tabachnick and Fidell 2013).

  1. Third, p.5, line 162, section 2.6 (line 168-171 revised manuscript). Pre- and post-test battery: It’s unclear to me why the experiment proceeded only at negative answers to the two latter questions. Should sleep quality, 24-hour physical activity, and consumption of drinks be considered?

Reply: We have rephrased this paragraph for better clarity (lines 168-171 in revised manuscript).  

  1. Finally, the authors introduced some abbreviations. They should be introduced for the first time they occur and should not be introduced redundantly. For example, "musculoskeletal disorders (MSD)" appears twice on line 34 and line 54.

Reply: Thank you for pointing out these formatting errors. We have edited the text accordingly (see manuscript, line 53 in revised manuscript).

Round 2

Reviewer 1 Report

After including all the requested modifications, the article has a greater consistency

Reviewer 4 Report

Thank you, the authors, for your responses and the revision. However, the following two questions were not clearly addressed.

First, thanks for adding descriptions about the environmental factors (lines 102-105). Reproducibility is critical in the scientific method. Values should be provided in the revision rather than a general description.

Second, the parameters that were used to calculate the sample size should be defined. The authors should assess whether their data violate the assumptions for RM-ANOVA or not.